



**Carbonate System Parameters of an Algal-dominated Reef along West Maui**
Nancy G. Prouty[1], Kimberly K. Yates[2], Nathan Smiley[2], Chris Gallagher[3], Olivia Cheriton[1], and Curt
D. Storlazzi[1]
[1] U.S. Geological Survey, Coastal and Marine Geology, Pacific Coastal and Marine Science Center, Santa Cruz, CA 95060
[2] U.S. Geological Survey, Coastal and Marine Geology, St. Petersburg Coastal and Marine Science Center 600 4th Street
South, St. Petersburg, FL 33701
[3] University of California, Santa Cruz, Santa Cruz, CA 95060
*Correspondence to*: Nancy G. Prouty nprouty@usgs.gov



## Abstract

Constraining coral reef metabolism and carbon chemistry dynamics are fundamental for understanding
and predicting reef vulnerability to rising coastal $CO_2$ concentrations and decreasing seawater pH.
However, few studies exist along reefs occupying densely inhabited shorelines with known input from
land-based sources of pollution. The shallow coral reefs off Kahekili, West Maui, are exposed to
nutrient-enriched, low-pH submarine groundwater discharge (SGD) and are particularly vulnerable to
the compounding stressors from land-based sources of pollution and lower seawater pH.  To constrain
the carbonate chemistry system, nutrients and carbonate chemistry were measured along the Kahekili
reef flat every 4 h over a 6-d sampling period in March 2016. Abiotic process - primarily SGD fluxes -
controlled the carbonate chemistry adjacent to the primary SGD vent site, with nutrient-laden
freshwater decreasing pH levels and favoring undersaturated aragonite saturation ($\Omega_{arag}$) conditions. In
contrast, diurnal variability in the carbonate chemistry at other sites along the reef flat was driven by
reef community metabolism. Superimposed on the diurnal signal was a transition during the second
sampling period to a surplus of total alkalinity (TA) and dissolved inorganic carbon (DIC) compared to
ocean end-member TA and DIC measurements. A shift from net community production and
calcification to net respiration and carbonate dissolution was identified. This transition occurred during
a period of increased SGD-driven nutrient loading, lower wave height, and reduced current speeds.
This detailed study of carbon chemistry dynamics highlights the need to incorporate local effects of
nearshore oceanographic processes into predictions of coral reef vulnerability and resilience.

## 1. Introduction

Coral reefs provide critical shoreline protection and important ecosystem services, such as marine habitat, and support local economies through tourism, fishing, and recreation ( Hughes et al., 2003; Ferrario et al., 2014). However, coral reefs are being threatened by global climate change processes, such as increasing temperatures, ocean acidification (OA), and sea-level rise, and these effects are often compounded by local stressors from over-fishing, sedimentation, coastal acidification, and land-based sources of pollution (Knowlton and Jackson, 2008). Isolating the effects of these stressors is difficult without establishing the biological and physical controls on community calcification and production. This is particularly challenging for coral reefs adjacent to densely inhabited shorelines, where freshwater fluxes can deliver excess nutrients, leading to eutrophication and coastal acidification, outbreaks of harmful algal blooms (Anderson et al., 2002), and decreased coral abundance and diversity (Fabricius, 2005;Lapointe et al., 2005). In many cases, eutrophication can alter ecosystem function and structure by shifting reefs from coral- to algae-dominated (Howarth et al.,



2000; Andrefouet et al., 2002; Hughes et al., 2007). Changes in community structure can have
profound impacts on coral reef metabolism and reef carbon chemistry dynamics, which are ultimately
linked to reef health, and the ability to predict future responses to rising $pCO_2$ levels (Andersson and
Gledhill, 2013). Understanding the local drivers of ecosystem function and reef community
metabolism is critical for gauging the susceptibility of the reef ecosystem to future changes in ocean
chemistry.

Numerous efforts have been conducted along west Maui, Hawaii, USA, to characterize and quantify
submarine groundwater discharge (SGD) and associated nutrient input (Dailer et al., 2010;Dailer et al.,
2012;Glenn et al., 2013; Swarzenski et al., 2013; Swarzenski et al., 2016). Until this study, however,
no field-based measurements of carbonate system parameters were available from the reefs in this area.
The carbonate chemistry system is sensitive to changes in photosynthesis, respiration, calcification,
and dissolution, and can be characterized by measuring total alkalinity (TA), dissolved inorganic
carbon (DIC), pH, $pCO_2$, nutrients, salinity, and temperature. Analysis of these parameters yields
valuable information on ratios of net community calcification and production, and can be used to
identify biological and physical drivers of reef health and ecosystem function (Silverman et al., 2007;
Shamberger et al., 2011; Lantz et al., 2014; Albright et al., 2015; Muehllehner et al., 2016; DeCarlo et
al., 2017).  Here, we present high temporal-resolution, in-situ measurements of carbonate chemistry
dynamics collected from the shallow coral reef off Kahekili in Kaanapali, west Maui, Hawaii, USA
(Fig. 1), with the aim of assessing the environmental controls on carbon metabolism (photosynthesis
and respiration, calcification and dissolution), and evaluating reef community performance and
function. This is particularly important given growing concern that coastal and ocean acidification may
shift reef ecosystems from net calcification to net dissolution by the mid to end of the century (
Silverman et al., 2009; Andersson and Gledhill, 2013) with an overall reduction in calcification rates
and increase in dissolution rates ( Shamberger et al., 2011; Shaw et al., 2012; Bernstein et al., 2016)
that can contribute to reef collapse (Yates et al., 2017).

The health of many of Maui's coral reefs has been declining rapidly (Rodgers et al., 2015), with recent
coral bleaching events leading to increased coral mortality (Sparks et al., 2016). The decline in coral
cover along the shallow coral reef at Kahekili has been observed for decades (Wiltse, 1996; Ross et al.,
2012), along with a history of macro-algal blooms (Smith et al., 2005). The shift in benthic cover from
abundant corals to turf- or macro-algae (primarily *Ulva fasciata*) and increased rates of coral
bioerosion has been linked to input of nutrient-rich water via wastewater injection wells (Dailer et al.,





2010;Dailer et al., 2012; Prouty et al. 2017a). Treated wastewater is injected through these wells into
groundwater that flows toward the coast where it emerges on the reef through a network of small seeps
and vents (Glenn et al., 2013;Swarzenski et al., 2016). Changes in coastal water quality observed off
west Maui can impact the balance of production of $CaCO_3$ skeletons by plants and animals on the reef,
cementation of sand and rubble, and $CaCO_3$ breakdown and removal that occurs through bioerosion,
dissolution, and offshore transport. Here, a high-resolution seawater sampling study was conducted to
constrain the carbonate chemistry system and evaluate the biological and physical processes altering
reef health along the shallow coral reef at Kahekili. This study represents the first characterization of
diurnal and multi-day variability of coral reef carbonate chemistry along a tropical fringing reef
adjacent to a densely inhabited shoreline with known input from land-based sources of pollution, and
identifies the controls on carbon metabolism. Ultimately, understanding carbonate system dynamics is
essential for managing compounding effects from local stressors.

**2. Methods**
*2.1 Study Site*
The benthic habitat along the shallow reef at Kahekili in Kaanapali, West Maui (Fig. 1) consists of
aggregate reef, patch reef, pavement, reef rubble and spur and groove (Cochran et al., 2014), with
persistent current flow to the south (Storlazzi and Jaffe, 2008).  Only 51% of the hardbottom at
Kahekili is covered with at least 10% live coral (Cochran et al., 2014). The shallow fore reef
experiences algae blooms, in response to inputs of nutrient-rich water via wastewater injection wells
(Dailer et al., 2010;Dailer et al., 2012).  Groundwater inputs occur from both natural sources (rainfall
and natural infiltration) and from artificial recharge (irrigation and anthropogenic wastewater).  The
inland Wailuku Basalt, consisting of a band of unconsolidated sediment along the coast, and a small
outcrop of Lahaina Volcanics, dominates the geology of the area surrounding the study site, controlling
the flow of groundwater.  Mean annual precipitation rates are up to 900 cm yr$^{-1}$ (Giambelluca et al.,
2013), with natural recharge the greatest in the interior mountains.

*2.2 Field Sampling*
Two intensive sampling periods were carried out during the 6-d period between 16 to 24 March 2016.
Seawater nutrients and carbonate chemistry variables were collected every 4 h during each sampling
period from the primary vent site and in adjacent coastal waters along the shallow reef at Kahekili (Fig.
1). The first sampling period was from 15:00 on 16 March 2016 to 15:00 on 19 March 2016, and the
second sampling period was from 15:00 on 21 March 2016 to 11:00 on 24 March 2016 (all reported



times in local [HST]). There were five sampling sites: two shallow (<1.5 m) sites (S1 and S2) located
approximately 10 m offshore, two deeper (5 m) sites (S3 and S4) located approximately 115 m
offshore, and a shallow site located approximately 20 m offshore and adjacent to an active SGD vent
(vent site) (Glenn et al., 2013;Swarzenski et al., 2016). Sampling tubes (ranging from approximately
100 to 200 m in length) were installed at each site by affixing the tube to a concrete block located
approximately 20 cm above the seafloor, or by attaching the tubing directly to dead reef structure using
zip ties. Tube intakes were fitted with a stainless steel screen cap to prevent uptake of large
particulates. The remaining length of each tube was positioned along the seafloor to the adjacent beach
by weighting the tube with a 1 m piece of chain, or by weaving the tube through dead reef structure
approximately every 20 m. The tube outflow ends were labeled for each sampling site, bundled in a
common location, and located above the high water line on the beach for sampling access. A peristaltic
pump was used to pump seawater from the seafloor. Sampling tubes were flushed for a minimum of 20
minutes to remove residual seawater before collecting data and water samples. Temperature (±
0.01°C), salinity (± 0.01), and dissolved oxygen (±0.1 mg L$^{-1}$) of water samples were measured using a
YSI ProPlus multimeter that was calibrated daily. However, due to temperature change during water
transit time within the sampling tube, in-situ temperatures were also recorded from Solonist CTD
Divers installed at the intake of each sampling tube. An upward-looking 2-MHz Nortek Aquadopp
acoustic Doppler profiler (ADP) was deployed at the southern deeper site (S4). The ADP sampled
waves at 2 Hz for 17 min every hour and currents at 1 Hz every 10 min in 1-m vertical bins from 1 m
above the seabed up to the ocean surface.

*2.3 Seawater Analyses*
Samples for dissolved nutrients ($NH_4^+$, Si, $PO_4^{3-}$, and [$NO_3^-+NO_2^-$]) were collected in duplicate by
filtering water with an in-line 0.45-μm filter and 0.20-μm syringe filter, and were kept frozen until
analysis. Nutrients were analyzed at the Woods Hole Oceanographic Institution's nutrient laboratory
and University of California at Santa Barbara's Marine Science Institute Analytical Laboratory via
flow injection analysis for $NH_4^+$, Si, $PO_4^{3-}$, and [$NO_3^-+ NO_2^-$], with precisions of 0.6-3.0%, 0.6-0.8%,
0.9-1.3%, and 0.3%-1.0% relative standard deviations, respectively. Select samples were collected and
analyzed for nitrate isotope ($\delta^{15}N$ and $\delta^{18}O$) analyses at the University of California at Santa Cruz
using the chemical reduction method (McIlvin and Altabet, 2005;Ryabenko et al., 2009) and
University of California at Davis' Stable Isotope Facilities using the denitrifier method (Sigman et al.,
2001). The isotope analysis was conducted using a Thermo Finnigan MAT 252 coupled with a
GasBench II interface; isotope values are presented in per mil (‰) with respect to AIR for $\delta^{15}N$ and





VSMO for $\delta^{18}$O with a precision of 0.3-0.4‰ and 0.5-0.6‰ for $\delta^{15}$N-nitrate and $\delta^{18}$O-nitrate,
respectively.

Seawater samples for determining carbonate chemistry variables (pH on the total scale, TA, and DIC)
were collected from the 5 sampling sites using a peristaltic pump and pressure filtering seawater
through a 0.45-μm filter. Samples for pH (±0.005) were filtered into 30-mL optical glass cells and
analyzed within 1 hr of collection using spectrophotometric methods (Zhang and Byrne, 1996), an
Ocean Optics USB2000 spectrometer, and thymol blue indicator dye. Samples for TA (±1 μmol kg$^{-1}$)
and DIC (±2 μmol kg$^{-1}$) were filtered into 300-ml borosilicate glass bottles, preserved by adding 100
μL saturated HgCl$_2$ solution and pressure sealed with ground glass stoppers coated with Apiezon
grease. TA samples were analyzed using spectrophotometric methods of (Yao and Byrne, 1998) with
an Ocean Optics USB2000 spectrometer and bromocresol purple indicator dye. DIC samples were
analyzed using a UIC carbon coulometer model CM5014 and CM5130 acidification module fitted with
a sulfide scrubber, and methods of (Dickson et al., 2007). In-situ temperatures recorded from Solonist
CTD Divers were reported and used to temperature-correct pH and perform CO2SYS calculations as
described below.

Certified reference materials (CRM) for TA and DIC analyses were from the Marine Physical
Laboratory of Scripps Institution of Oceanography (Dickson et al., 2007).  Duplicate or triplicate
analyses were performed on at least 10% of samples, yielding a mean precision of ~1 μmol kg$^{-1}$ and ~2
μmol kg$^{-1}$ for TA and DIC analyses, respectively. The full seawater $CO_2$ system was calculated with
measured salinity, temperature, nutrients (phosphate and silicate), TA, DIC, and pH data using an
Excel Workbook Macro translation of the original CO2SYS program (Pierrot et al., 2006).  Given the
enriched nutrient setting of the study site, TA values were nutrient corrected in CO2SYS (Dickson,
1981).  The aragonite saturation state and $p$CO$_2$ are reported based on TA-pH pairs, with dissociation
constants $K_1$ and $K_2$ from (Mehrbach et al., 1973) refit by (Dickson and Millero, 1987) and KSO4 from
(Dickson, 1990).  The TA and DIC values were normalized to salinity (by multiplying by a factor of
35/S, where S is the measured salinity value) to account for variations in TA and DIC driven by
evaporation and/or precipitation (Friis et al., 2003) and are reported as $n$TA and $n$DIC as previously
established in reef geochemical surveys (e.g., Suzuki and Kawahata, 2003; Yates et al., 2014;
Muehllehner et al., 2016). However at the vent site the TA and DIC data was not normalized to salinity
given the contribution of TA and DIC from SGD.



## 3. Results

### 3.1 Submarine Groundwater Endmember

The magnitude of change and absolute values in the carbonate chemistry, nutrients, and salinity were greatest at the primary vent site relative to the four sites along the reef. The salinity ranged from 10.64 to 36.72 over the 6-d period (Fig. 2A), with the most dramatic decrease in salinity on March 22[nd] when salinity decreased from 32.45 to 12.47 within 4 hr. The reduction in salinity was sustained over a 32-hr period. A rapid change was also observed in the pH, DO, TA, DIC, and nutrient concentrations (Fig. 2). For example, nitrate concentrations at the vent site ranged from 0.45 to over 70 $\mu$mol L$^{-1}$, with an average nitrate concentration of 117 (SD 0.09) $\mu$mol L$^{-1}$ measured directly from the discharging seep water. The $\Omega_{arag}$ values decreased to less than 1 and $pCO_2$ values increased to 2000 $\mu$atm when salinity values dropped to less than 15 (Fig. 2D). No diurnal pattern was detected in the seawater carbonate chemistry at this site. Instead, these results are consistent with earlier work documenting lower pH, nutrient enriched freshwater endmember values tightly coupled to SGD ( Swarzenski et al., 2012; Glenn et al., 2013;Swarzenski et al., 2016).

### 3.2 Reef Flat

In contrast to the vent site, the overall magnitude of carbonate chemistry variation at the other four sites along the reef flat was less, and the signal was coherent among these sites. This coherency is captured in the pH time series (Fig. 3B), where the pH data from the four sites were significantly ($p<0.05$) positively correlated with each other (with $r \sim 0.5$). The lowest salinity value along the reef flat was 33.51, indicating minimal freshwater influence on reef flat salinity. As a result, the carbonate system parameters measured along the reef were non-linear with respect to salinity, instead a diurnal pattern dominated the signal (Fig. 3). Lowest pH values occurred around midnight (23:00); and highest pH values occurred in the afternoon (~14:00-15:00). This diurnal pattern was also apparent in the DIC data, with lowest values in the afternoon and increasing around midnight, with a cubic spline fit (Press et al., 1988) highlighting diurnal cycle from all four sites along the reef flat. Likewise, the diurnal signal was identifiable in the $\Omega_{arag}$ and $pCO_2$ time-series, with $\Omega_{arag}$ values increasing and $pCO_2$ decreasing during the mid-day hours (Fig. 3). The diurnal signal in the $n$TA time-series was similar to the signal for $n$DIC. At the shallow (<5 m) sites, pH and DO covaried ($r^2$=0.43-0.87; $p<0.001$). The range in pH and $\Omega_{arag}$ was largest at the shallow sites; however, the average values were similar along the reef, 3.02 to 3.06 and 8.00 to 8.01, respectively, and were elevated relative to the average values recorded at the vent site, 7.85 (SD 0.17) and 2.28 (SD 0.81) for pH and $\Omega_{arag}$, respectively (Prouty et al., 2017b). No diurnal pattern was observed for the nutrient data; however, there was an offshore



gradient in nutrient concentrations with enriched nutrients at the shallow sites compared to the deeper
sites. Nutrient concentrations (Si, $PO_4^{3-}$, and $NO_3^-$) from the two shallow sites were statistically greater
than the two deeper sites according to pairwise multi-comparison one-way ANOVA with a *post hoc*
Tukey HSD ($p>0.05$).  For example average nitrate concentrations at the two shallow sites were 0.71
(SD 0.35) and 0.41 (0.18 SD) compared to 0.17 (SD 0.10) and 0.19 (SD 0.11) $\mu$mol $L^{-1}$. Deficits and
surpluses of $n$TA and $n$DIC, with respect to open ocean conditions, were calculated as $\Delta n$TA and
$\Delta n$DIC using values from Station HOT (Dore et al., 2009), located approximately 250 km offshore.
The $\Delta n$TA values ranged from -332 $\mu$mol $kg^{-1}$ to 85 $\mu$mol $kg^{-1}$ and -171 $\mu$mol $kg^{-1}$ to 141 $\mu$mol $kg^{-1}$
$\Delta n$DIC. The standard error of difference ($SE_{dif}$) was calculated for $\Delta n$TA and $\Delta n$DIC values to evaluate
whether the deficits and surpluses of $n$TA and $n$DIC were significant. Histogram plots reveal statistical
($p=0.05$; critical t value of 1.68; df=37) deficits and surpluses as well as differences between the first
and second half of the sampling period (Fig. 4). Results show a shift from a deficit in $\Delta n$TA to a
surplus in $\Delta n$TA at all stations, as well as a shift from a deficit in $\Delta n$DIC to a surplus in $\Delta n$DIC,
suggesting a shift in the second sampling period from net $CaCO_3$ production to net $CaCO_3$ dissolution,
and from net photosynthesis to net respiration. This change was most distinct at the two shallow sites.
The $n$TA and $n$DIC values from the second sampling period were also enriched relative to a range of
values reported from nearshore Oahu sites (Drupp et al., 2013).

## 4. Discussion

The diurnal pattern observed at the four sampling sites along the reef flat is typical of a reef
environment where biotic processes involving coral reef community metabolism (e.g.,
respiration/photosynthesis and calcification/dissolution) dominate the carbonate chemistry system
(e.g., Smith, 1973). The non-linear relationship between salinity and carbonate chemistry parameters
further supports the notion that biotic processes are driving carbonate chemistry variability along the
reef flat (Millero et al., 1998; Ianson et al., 2003). The lower amplitude $n$TA diurnal signal supports
previous observations that the region was algal-dominated (Smith et al., 2005). In this case, the lower
biomass of calcifying organisms leads to conditions that favor respiration-photosynthesis processes
relative to calcification-dissolution (Jokiel et al., 2014). Elevated pH values during mid-day, coincident
with elevated sea surface temperature (SST) and peak solar irradiance, are consistent with maximum
photosynthetic activity. DIC decreased during the day due to photosynthesis, whereas at nighttime, pH
decreased and DIC increased in response to respiration (Fig. 3).  This pattern is in stark contrast to the
primary vent site where no diurnal pattern was observed, and abiotic controls on the carbonate system
dynamics explain the strong linear relation to salinity. Variability at the vent site is driven by SGD





rates, which are elevated during low tide when hydraulic gradients are the steepest (Dimova et al.,
2012;Swarzenski et al., 2016).

To further understand the temporal variability in carbonate chemistry over the 6-d sampling period
along the reef flat, diagrams of $n$TA versus $n$DIC were plotted according to Zeebe and Wolf-Gladrow,
(2001), along with vectors indicating theoretical effects of organic and inorganic carbon metabolism on
seawater chemistry (Kawahata et al., 1997;Suzuki and Kawahata, 2003) (Fig 5). Diagrams of $n$TA-
$n$DIC indicate the dominance of net community production (NCP) and net community calcification
(NCC) during the first sampling period (16-19 March). The slope values of the $n$DIC-$n$TA plots were
used to calculate ratios of NCC:NEP (Table 1) using methods of Suzuki and Kawahata (2003). In the
absence of reliable water mass residence time, ratios were used rather than metabolic rates. The
NCC:NEP ratios for the first sampling period ranged from 0.50 to 0.87 indicate that both calcification
and photosynthesis contributed to variability in carbonate system parameters with photosynthesis as
the dominant processes in all cases. This pattern was observed at all four sites along the reef flat. In
comparison, a shift occurred after the first sampling period. Elevated $n$DIC and $n$TA values during 21-
24 March indicate a shift to primarily respiration and dissolution in the $n$TA-$n$DIC diagrams (Fig. 5).
At the shallow sites, S1 and S2 (Fig. 5A and B), the NCC:NCP ratios were 0.56 and 0.39 (Table 1),
respectively, indicating primarily net respiration at these locations. On Heron Island for example, high
organic production results in NCC:NCP ratios between 0.25 and 0.29 (McMahon et al., 2013; Albright
et al., 2015). Dissolution and respiration contributed nearly equally with NCC:NCP ratios near 1.0 at
sites S3 and S4 located further offshore. Rather than reflecting an artifact of the salinity normalization,
given the non-linear relation of DIC and TA to salinity along the reef flat, this shift is interpreted as a
reef community response. As shown in Figures 4 and 5, this change captures a shift from a reef
community dominated by calcification to one dominated by respiration and dissolution.

The shift from net photosynthesis (P) to net respiration (R) as captured in the Δ$n$DIC histogram plots
(Fig. 4), suggests that the coral-algal association consumed more energy than it produced during the
second sampling period. As a proxy for autotrophic capacity, the change in P:R ratio may reflect an
increase in coral heterotrophic feeding relative to autotrophic feeding (Coles and Jokiel, 1977;Hughes
and Grottoli, 2013). Typically, stored lipid reserves in the tissue are utilized when the stable symbiotic
environment is disturbed (e.g., Szmant and Gassman, 1990; Ainsworth et al., 2008). Although short-
lived, thermally-induced bleaching has been linked to depletion of coral lipid reserves (e.g., Hughes
and Grottoli, 2013), excess nutrient loading can also shift the stability of the coral-algae symbiosis,





thereby reducing stored tissue reserves (Wooldridge, 2016). According to Glenn et al. (2013), up to 11
$m^3$ $d^{-1}$ of dissolved inorganic nitrogen are discharged onto the West Maui reef as the result of receiving
and treating over 15,000 $m^3$ $d^{-1}$ of sewage. Using a SGD flux rate of 87 cm $d^{-1}$ at the primary seep site
(Swarzenski et al., 2016), and SGD nitrate end-member concentration of 117 μmol $L^{-1}$ (Prouty et al.,
2017b), the nitrate flux from the primary vent site is 712 mol $d^{-1}$, clearly demonstrating excess nutrient
loading.  As described above, an offshore gradient in nutrient concentrations was observed with
enriched nutrients at the shallow sites compared to the deeper sites, consistent with a decrease in coral
$δ^{15}N$ values away from the vent (Prouty et al., 2017a). Coral tissue thickness was also negatively
correlated to coral tissue $δ^{15}N$ values ($r = -0.66$; $p = 0.08$), with the latter serving as a proxy for
nutrient loading in alga samples along the reef flat (Dailer et al., 2010). It is possible that a reduction in
coral tissue reflects preferential heterotrophic feeding under high nutrient loading, with nutrient
enrichment by sewage effluent increasing primary production and biomass in the water column (e.g.,
Smith et al., 1981; Pastorok and Bilyard, 1985). While assessing the impacts of nutrient loading on
coral physiology may be long term and subtle in some cases, results from our study highlight the
potential short-term impacts of nutrification on the short term.

Identifying the exact mechanism(s) responsible for driving this shift is difficult given the complexity of
the reef system.  Possible explanations include warmer SSTs, suspension of organic matter, as well as
secondary effects of nutrification from contaminated SGD (D'Angelo and Wiedenmann, 2014).
Given that microbial communities rapidly take up inorganic nutrients (Furnas et al., 2005), there could
be increased respiration as a result of increased microbial remineralization of organic matter in the
nutrient-loaded environment.  In other words, enhanced SGD- driven nutrient fluxes during the second
sampling period could have increased microbial growth and remineralization, shifting the reef
community metabolism, as captured in a shift in the carbonate chemistry system.  In addition to
community metabolism, local oceanographic effects such as the wind and wave regime can also drive
carbonate chemistry by altering air-sea exchange and water mass residence times. During the first
sampling period, the wave height increased from 0.4 m to 1.6 m over the first 2 d and mean current
speeds were 1.6 cm $s^{-1}$ (Fig. S1). In comparison, during the second sampling period, wave height
declined to less than 0.4 m and mean current speeds were 1.0 cm $s^{-1}$. Together, the reduced wave
height and reduced wind speeds favor slower release of $CO_2$ generated by calcification and respiration
processes from the water column (Massaro et al., 2012), resulting in higher $pCO_2$ and lower pH.

Despite being situated in an oligotrophic region with naturally occurring, low nutrient concentrations,



anthropogenic nutrient loading to coastal waters via sustained SGD is driving nearshore eutrophication
( Dailer et al., 2010; Dailer et al., 2012; Bishop et al., 2015; Amato et al., 2016; Fackrell et al., 2016),
with algal $\delta^{15}$N signatures at Kahekili Beach Park indicative of wastewater effluent (Dailer et al.,
2010;Dailer et al., 2012). In response, there has been a shift in benthic cover from abundant corals to
turf- or macro-algae over the last two decades.  Areas of discrete coral cover loss up to 100% along the
shallow coral reef at Kahekili have been observed for decades ( Wiltse, 1996; Ross et al., 2012), with a
history of macro-algal blooms (Smith et al., 2005).   More recently, Prouty et al. (2017a) found
accelerated nutrient driven-bioerosion from coral cores collected along the Kahekili reef flat in
response to land-based sources of nutrients.  This is consistent with earlier work showing nutrification-
mediated increase in plankton loads can trigger increases in filter feeders and bioeroders that endanger
reef structure integrity (e.g., Fabricius et al., 2012).  Eutrophication from nutrient enriched SGD may
contribute to an already compromised carbonate system (i.e., reduced pH and $\Omega_{arag}$) by increasing net
respiration and remineralization of excess organic matter, and increasing bioerosion.   Therefore,
secondary effects of nutrient-driven increase in phytoplankton biomass and decomposing organic
matter are also important considerations for coral reef management (D'Angelo and Wiedenmann,

2014).


As discussed above, SGD rates are elevated during low tide when the relative pressure head between
terrestrial groundwater and the oceanic water column is greatest (Dimova et al., 2012;Swarzenski et
al., 2016). Relative SGD is greater in the shallows close to shore where the tidal height is larger
relative to the depth of the water column. Higher islands, therefore, have the potential for not only
greater orographic rainfall and thus submarine groundwater recharge, but also greater potential
pressure head and thus enhanced SGD- driven nutrient fluxes. There is also greater potential for
enriched nutrient sources and reduced water quality with fast-growing population and development
(Amato et al., 2016;Fackrell et al., 2016). Thus, SGD represents a key vector of nutrient loading in
tropical, oligotrophic regions (e.g., Paytan et al., 2006). At the same time, closer to shore, current
speeds are generally slower resulting in longer water mass residence times (Storlazzi et al., 2006);
longer residence times would also be expected closer to the seabed, compared with upper water
column flows (Storlazzi and Jaffe, 2008). Together, these suggest that the resulting exposure (=
intensity x residence time) of coral reefs to nutrient-laden, low pH submarine groundwater is greater
for coral reefs closer to shore off high islands than along barrier reefs or on atolls. This heightened
vulnerability therefore needs to be taken into account when evaluating vulnerability of nearshore
fringing reefs to changes in carbonate chemistry system given evidence of nutrient driven-bioerosion



from land-based sources of pollution.


**5. Conclusion**
Field based measurements of carbonate chemistry variability were made along a shallow coral reef off
Kahekili, west Maui, and captured differences in the relative importance of inorganic and organic
carbon production over a 6-d period in March 2016. Submarine groundwater discharge fluxes
controlled the carbonate chemistry adjacent to the primary vent site, with nutrient-laden freshwater
decreasing the pH levels and favoring undersaturated $\Omega_{arag}$ conditions. In contrast, reef community
metabolism dominated the carbonate chemistry diurnal signal at sites along the reef flat. Superimposed
on the diurnal signal was a transition during the second sampling period, yielding a surplus of $n$TA and
$n$DIC compared to ocean endmember measurements indicating a shift from net photosynthesis and
calcification to net respiration and carbonate dissolution. This shift could be interpreted as a direct
response to increased nutrient loading, and subsequent enhancement of organic matter
remineralization.  Predictions of reef response to elevated $p$CO$_2$ levels assume reef water tracks open-
ocean pH, however local effects are equally important (e.g., Cyronak et al., 2013), particularly along
densely-inhabited shorelines with known input from land-based sources of pollution. Building on
previous work documenting the input of nutrient laden, low-pH freshwater to the reefs off Kahekili,
results presented here offer a first glimpse into how anthropogenic-driven eutrophication might add an
additional stressor to thresholds tipping the balance between net carbonate accretion and net carbonate
dissolution, thus altering carbonate system dynamics.

**Author contribution**
NGP and KKY designed the experiments and NGP, KKY, NS and CG carried them out. NGP and
KKY completed the chemical measurements and OC compiled the oceanographic data. NGP prepared
the manuscript with contributions from all co-authors.

The authors declare that they have no conflict of interest.

**Acknowledgements**
This research was carried out as part of the US Geological Survey's Coral Reefs Project in an effort to
better understand the effects of geologic and oceanographic processes on coral reef systems in the
USA and its trust territories, and was supported by the USGS Coastal and Marine Geology Program.
The authors gratefully acknowledge the vital partnership and expert logistics support provided by the
State of Hawaii Division of Aquatic Resources. We thank T. DeCarlo (UWA), H. Barkley (NOAA) for



reviews, and A. Cohen (WHOI) for helpful discussions, K.R. Pietro and K. Hoering (WHOI), C.
Moore (USGS), and G. Paradis (UCSB) analytical assistance, M. Dailer (U. Hawaii) for field
assistance. The use of trade names is for descriptive purposes only and does not imply endorsement by
the U.S. Government.

**Figure Captions**
**Figure 1.** Location map of the island of Maui, Hawaii, USA, and the study area along west Maui.
Bathymetric map (5-m contours) of study area showing seawater sampling locations (blue closed
circle) along Kahekili Beach Park, and the primary seep site (blue open circle) superimposed on
distribution of percent coral cover versus sand.

**Figure 2** Results of time-series of seawater chemistry variables over a 6-d period collected from
bottom water near the seep site on the nearshore reef every 4 hr. (A) Salinity, (B) dissolved nutrient
(nitrate+nitrite, phosphate, and silicate) concentrations ($\mu$mol L$^{-1}$), and nitrate stable nitrogen isotopes
($\delta^{15}$N-nitrate; ‰), (C) total alkalinity (TA) and dissolved inorganic carbon (DIC) ($\mu$mol kg$^{-1}$), (D)
calculated carbonate parameters for aragonite saturation state ($\Omega_{arag}$), and $pCO_2$ ($\mu$atm; inverted) based
on TA-pH pairwise and measured salinity, temperature, nutrients (phosphate and silicate) data, (E)
dissolved oxygen (DO; mg L$^{-1}$), and (F) temperature corrected pH (total scale). End-of-century
projections according to IPCC-AR5 RCP8.5 "business as usual" scenario for pH (reduction by 0.4
units), $\Omega_{arag}$ (2.0; blue dashed), and $pCO_2$ (750 $\mu$atm; red dashed).

**Figure 3** Carbonate chemistry parameters and sea surface temperature (SST) composite from S1, S2,
S3 and S3 along the shallow reef flat of Kahekili, Maui and cubic spline fits highlighting diurnal cycle
for the first (16-19 March 2016; solid line) and second (21-24 March 2016; dashed line) sampling
period for (A) Temperature, (B) pH, (C) $n$DIC and (D) $n$TA ($\mu$mol kg$^{-1}$), (E) $\Omega_{arag}$ and (F) $pCO_2$
($\mu$atm).

**Figure 4** Histogram $\Delta n$TA and $\Delta n$DIC capturing deficits and surpluses of $n$TA and $n$DIC with respect
to open ocean conditions. Overall a transition from net CaCO$_3$ production to net CaCO$_3$ dissolution
and net photosynthesis to net respiration occurred between the first (16-19 March 2016; blue) and
second (21-24 March 2016; red) sampling period for the shallow sampling sites (A)-(B) S1 and (C)-
(D) S2, and the two deeper sites (E)-(F) S3, and (G-H) S4. Statistical ($p$=0.05) deficit and surplus
values (±) for $\Delta n$TA and $\Delta n$TA shown in parentheses.




**Figure 5** Seawater carbonate chemistry system along the reef flat off Kahekili as a function of $n$DIC
and $n$TA for the shallow sampling sites (A). S1 and (B) S2, and two deeper sites (C) S3, and (D) S4 for
the first (blue) and second (red) sampling periods and their respective slopes (solid lines) of $n$DIC and
$n$TA (Table 1) and theoretical slope (dashed lines) given the predicted effects of photosynthesis,
respiration, calcification, and dissolution as shown in (E) and the respective change in net community
calcification (NCC) and net community production (NCP).

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




| Site | $n$TA-$n$DIC Slope | NCC:NCP | $r^2$ |
|------|------|------|------|
| *16-19 March 2016* | | | |
| S1 | 0.88 | 0.78 | 0.94 |
| S2 | 0.67 | 0.50 | 0.75 |
| S3 | 0.93 | 0.88 | 0.89 |
| S4 | 0.93 | 0.87 | 0.92 |
| | | | |
| *21-24 March 2016* | | | |
| S1 | 0.72 | 0.56 | 0.78 |
| S2 | 0.56 | 0.39 | 0.77 |
| S3 | 0.99 | 0.98 | 0.95 |
| S4 | 1.04 | 1.08 | 0.94 |


**Table 1**
Slope of salinity normalized total alkalinity ($n$TA): salinity normalized dissolved inorganic carbon
(DIC), net community calcification: net community production ratio (NCC:NCP=2ΔDIC/ΔTA-1)
(Suzuki and Kawahata, 2003) and correlation coefficients ($r^2$).





Figure 1

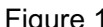



Figure 2





Figure 3







Figure 4





Figure 5

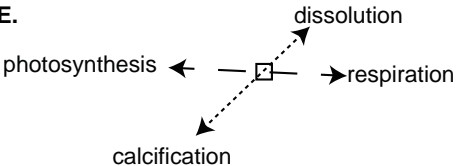