# Peer review of "Carbonate System Parameters of an Algal-dominated Reef along West Maui"

_Biogeosciences, 2018_

## Referee Comment (RC1) · Anonymous Referee #1 · 30 Jan 2018

"General Comments" Overall, this is a very nice paper that is scientifically sound and contains very few technical errors. The authors measured seawater carbonate chemistry and nutrients at shallow fringing reefs around a submarine groundwater discharge site to show what's driving chemical variability at these shallow sites with local anthropogenic stressors. They showed that chemistry (salinity, carbonate chemistry, DO) was highly variable at the vent site and driven by SGD while most parameters had a diurnal signal on the reef due to benthic metabolism. They also showed that areas closest to the vent site experienced a shift in NCC and NCP that may relate to nutrients being discharged from the vent. This study is scientifically sound and addresses a critical knowledge gap of understanding natural drivers of seawater carbonate chemistry variability on reefs, which must be understood in order to predict the effects of long-term

anthropogenic ocean acidification on reefs. My main critique of this paper is clarification of the terminology in order to more accurately draw conclusions about benthic metabolism from the available data they collected.

"Specific Comments" Introduction This manuscript gave a nice introduction to the research and sets the reader up for understanding and interpreting the results. However, the research goals were stated twice and therefore seemed repetitive. Typically, the research objectives are listed near the end of the introduction. It also was difficult to tie different parts of the introduction together, but hopefully the specific comments below will help address the flow:

Lines 36-37: Need to define OA versus coastal acidification. I assume the authors are referring to OA as a long-term anthropogenic effect owing to uptake of CO2 while coastal acidification refers to natural processes.

Lines 36-40: These are nice introductions to stressors on reefs and community metabolism; however, the tie between the two is not clear as presently written. Perhaps consider adding a transition between these two statements stating how these stressors are affecting reefs (e.g. decreased calcification, increased dissolution, etc.) and then go into community metabolism

Lines 52-53: Again, I felt like this was an abrupt transition. Could add "which may influence reef metabolism and community composition" at the end of the sentence.

Line 57: add "calcium carbonate (CaCO3)" in front of dissolution

Methods Lines 95-96: What is the other 90% of cover where there is 10% live coral cover? What is the community composition of the other 49% of hard-bottom area? This would help with interpretation of results and DIC/TA slopes as this relates to the community composition (corals vs algae vs sand, etc. See Page et al 2016 for reference on community composition influence on seawater carbonate chemistry.)

Lines 111-114: What was the approximate depth of the vent site? This would be valuable information in interpreting the variability (measured as daily range) of chemistry since depth can be such a strong control (Falter et al 2013).

I do wonder about any algae, bacterial films, etc. that may have grown on the inside of the tubing and possibly influenced carbonate chemistry and nutrients. Were there any tests (e.g. sampling carbonate chemistry near the intake and at the outtake) to assess whether the tubing was clean throughout the entire field study?

Thanks for providing the approximate precision of the TA and DIC measurements. It would be great to see the actual precision and accuracy (as mean plus/minus sd) of pH, TA, and DIC though.

What carbonate parameters are actually used for the pCO2 and saturation state calculations? This was unclear to me at this point of the manuscript but later it states they were calculated from TA-pH pairing. Please clarify in the methods.

What kind of filters were used for nutrients and carbonate chemistry sampling? Some filters may alter the values due to reactions between seawater and the material of the filters.

Results The results are very well-written. Just one clarification:

Line 215: What range of dates were used to calculate values for the open ocean site?

Discussion Line 240: Respiration also occurs during the day, not just at night. Could state "net respiration" rather than just "respiration"

Lines 249-251: How can both NCP and NCC dominate? It's unclear whether the authors are trying to say they are more balanced compared to the 2nd sampling or whether they mean "net photosynthesis (+NCP)" and "net calcification (+NCC)."

Lines 252-254: Please use NEC/NEP or NCC/NCP to maintain consistency with the scales used in this study. Also, please define these terms either here or in the introduction.

Lines 254-255: These should be "net calcification" and "net photosynthesis" to more accurately reflect what is actually measured. NCC and NCP can indicate net processes (calcification-dissolution or photosynthesis-respiration).

Line 260: The lower NCC:NCP ratio only indicates dominance by organic carbon cycling (vs inorganic carbon cycling), not which process (photosynthesis, respiration, calcification, dissolution) is actually dominating.

Lines 260-262. This statement seems a little out of place and I'm not sure what point the authors are trying to convey. Why are the slopes in this study higher than Heron Island? Does this reflect differences in benthic community composition, ecosystem function, or a combination?

Line 262: Again, "net dissolution" and "net respiration" since actual rates are not measured using this methodology

Does the nitrate end member at the vent site vary temporally? I appreciate using the available data to show the SGD but wonder how closely it represents discharge during the time of this study.

Figures/Figure Captions Line 389: "seep site AND on the nearshore..."

Line 393: So were TA and pH used to calculate pCO2 and saturation state? This was not clear in the methods.

Figure 5: Please show error bars for the open ocean since this presumably represents a mean. NCC and NCP need to be defined either in the caption or text. In part E, these should all be shown as "net..." Rather than just showing the arrows for part E, could you put it on a TA/DIC plot? It can even be shown right on the plots for A-D. Given your discussion of the data, I personally would rather see the processes as small arrows on a subplot (or just in the corner of a plot) and then have dashed lines indicating the transitions between +NCC/-NCC and +NCP/-NCP. I think this would make it easier for the reader to go back and forth between the figure and discussion.

"Technical Corrections" Line 47: no comma necessary Line 97: no comma necessary Line 112: Is 115 a typo? Should it be 15? Lines 154 and 157: parentheses just around the year Line 297: no space in SGD-driven

———————————————————

---

## Referee Comment (RC2) · Anonymous Referee #2 · 21 Feb 2018

This is a very interesting and very well-written paper that will definitely be a nice contribution to the field. There are a few major and minor comments below that I feel need to be addressed prior to publication.

My biggest criticism is that the authors did not account for TA and DIC fluxes from the SGD itself. This is an important step to interpret how much of the delta TA or delta DIC is due to reef metabolism. The authors also need to add a data analysis section to the methods and state all their statistical approaches and programs used to analyze the data. The remaining comments are relatively minor.

Line 52: There are other carbonate data for Kahekili (see, Silbiger et al. 2017 Ecology), but it is extremely limited. This is by far the most comprehensive study at this site, but "no field-based measurements" is inaccurate.

[Figure]

Line 81: Change "plants" to calcifying algae

Line 85: This is the first at Kahekili, but not the first to constrain carbonate chemistry in response to SGD (see Richardson et al. 2017 L&O). I would remove this sentence.

Line 124: Put both accuracy and precision of the instruments.

Line 168: Why did you use the TA-pH pairs rather than the TA-DIC pairs for the omega calculations? TA-pH is fine, but TA-DIC has less error propagation for calculating omega and it seems that you have those data.

Line 171: It is not clear which TA, DIC values you are talking about here.

Add a data or statistical analysis section at the end of the methods and discuss how you analyzed your data here. What program did you use for your stats?

What were the TA values coming directly out of the seep?

When calculating delta TA and DIC, the SGD endpoint needs to be taken into account. SGD can have a dramatically different TA and DIC concentrations than seawater (see Nelson et al. 2015 Marine Chem). A good portion of the TA and DIC fluxes are thus likely due to SGD and the remainder after accounting for these fluxes are due to biological processes (e.g., calcification, dissolution, P,R). Examples of studies that have accounted for fluxes of TA and/or DIC from freshwater sources are Paquay et al 2007 Aquatic geochem or Richardson et al. 2017 L&O

Line 234: The TA amplitude could also be indicative of high dissolution rates or a biproduct of the TA flux from the SGD onto the reef.

Line 251: Put this information in the methods and explain how you did the calculation in addition to citing the paper.

Line 290: remove "on the short term" at the end of the sentence. There is no physiology data in this study, so this sentence is a bit of a stretch. It does however look at ecosystem functioning of reefs.

Line 297: add a citation after "environment."

In the discussion, it would be interesting if the authors compared their results to with other studies that also measured carbonate chemistry at SGD sites (e.g., Nelson et al. 2015 Marine Chem and Richardson et al. 2017). Are the patterns similar or different?

Figures: make the colors more contrasting in the figures so that people printing in black and white can see the differences.

---

## Author Comment (AC1) · 21 Mar 2018

Prouty et al. Anonymous Referee #1

R1: "General Comments" Overall, this is a very nice paper that is scientifically sound and contains very few technical errors. The authors measured seawater carbonate chemistry and nutrients at shallow fringing reefs around a submarine groundwater discharge site to show what's driving chemical variability at these shallow sites with local anthropogenic stressors. They showed that chemistry (salinity, carbonate chemistry, DO) was highly variable at the vent site and driven by SGD while most parameters had

a diurnal signal on the reef due to benthic metabolism. They also showed that areas closest to the vent site experienced a shift in NCC and NCP that may relate to nutrients being discharged from the vent. This study is scientifically sound and addresses a critical knowledge gap of understanding natural drivers of seawater carbonate chemistry variability on reefs, which must be understood in order to predict the effects of long-term anthropogenic ocean acidification on reefs. My main critique of this paper is clarification of the terminology in order to more accurately draw conclusions about benthic metabolism from the available data they collected.

AR: We appreciate the valuable comments from Anonymous Referee #1 and believe we have conscientiously addressed the suggestions in the text and have detailed our responses below.

R1: "Specific Comments" Introduction This manuscript gave a nice introduction to the re- search and sets the reader up for understanding and interpreting the results. However, the research goals were stated twice and therefore seemed repetitive. Typically, the research objectives are listed near the end of the introduction. It also was difficult to tie different parts of the introduction together, but hopefully the specific comments below will help address the flow:

AR: We agree with the reviewer's critique regarding repetition in the introductory paragraph and we have omitted the research objectives stated earlier in the introductory paragraph.

R1: Lines 36-37: Need to define OA versus coastal acidification. I assume the authors are referring to OA as a long-term anthropogenic effect owing to uptake of $CO_2$ while coastal acidification refers to natural processes.

AR: As noted by the reviewer, it is important to distinguish between ocean acidification (OA) and coastal acidification. Ocean acidification is largely driven by the uptake of atmospheric anthropogenic $CO_2$ in oceanic waters (Orr et al., 2005) whereas coastal acidification is believe to be largely explained by processes such as contributions from

freshwater inflow, upwelling and/or eutrophication (Cai et al., 2011) whereby excess nutrient loading from human activities to coastal waters enhances respiratory processes that release CO2 and in turn increase coastal water acidity (e.g., see review by Strong et al., 2015). We have included a clarification statement in the revised manuscript in the Introduction section.

R1: Lines 36-40: These are nice introductions to stressors on reefs and community metabolism; however, the tie between the two is not clear as presently written. Perhaps consider adding a transition between these two statements stating how these stressors are affecting reefs (e.g. decreased calcification, increased dissolution, etc.) and then go into community metabolism.

AR: Per the reviewer's suggestion, we have modified these statements for greater clarity and have revised the introduction to include the following statement: "These stressors can lead to a decrease in reef health by removing grazing fish, decreasing calcification rates, and increasing nutrient and contaminant concentrations, thereby shifting the balance between reef accretion and bioerosion."

R1: Lines 52-53: Again, I felt like this was an abrupt transition. Could add "which may influence reef metabolism and community composition" at the end of the sentence. AR: Per the reviewer's suggestions we have modified this statement to include "...which may influence reef metabolism and community composition by changing coastal water quality."

R1: Line 57: add "calcium carbonate (CaCO3)" in front of dissolution

AR: Per the reviewer's suggestions we have inserted "calcium carbonate (CaCO3)" in front of dissolution

R1: Methods Lines 95-96: What is the other 90% of cover where there is 10% live coral cover? What is the community composition of the other 49% of hard-bottom area? This would help with interpretation of results and DIC/TA slopes as this relates

to the community composition (corals vs algae vs sand, etc. See Page et al 2016 for reference on community composition influence on seawater carbonate chemistry.)

AR: A detailed discussion of seafloor-bottom type can be found in Cochran et al. (2014). In brief, the remaining 49% of available hardbottom consists of aggregate reef, spur-and-groove, patch reefs, pavement, and reef rubble, which as the reviewer points can influence seawater carbonate chemistry (Page et al., 2017). In addition to live coral cover, Cochran et al. (2014) observed macroalgae, corraline algae, seagrass, and turf in the area mapped (5 km2 of sea floor from the shoreline to water depths of ∼30 m), however the sampling sites in our study were areas of live coral cover. We have included additional information in the Methods Section and reference to Page et al. (2017) in the Introduction Section.

R1: Lines 111-114: What was the approximate depth of the vent site? This would be valuable information in interpreting the variability (measured as daily range) of chemistry since depth can be such a strong control (Falter et al 2013).

AR: The vent site was located at a comparable depth to the two shallow (<1.5 m) sites (S1 and S2). We have inserted the water depth in the revised manuscript.

R1: I do wonder about any algae, bacterial films, etc. that may have grown on the inside of the tubing and possibly influenced carbonate chemistry and nutrients. Were there any tests (e.g. sampling carbonate chemistry near the intake and at the outtake) to assess whether the tubing was clean throughout the entire field study?

AR: Sampling tubes were flushed for a minimum of 20 minutes to remove residual seawater before collecting data and water samples. In addition, the tube intakes were fitted with a stainless steel screen cap to prevent uptake of large particulates. We also inspected the tubes upon extraction and found no significant algal growth. The revised manuscript includes this additional information.

R1: Thanks for providing the approximate precision of the TA and DIC measurements.

It would be great to see the actual precision and accuracy (as mean plus/minus sd) of pH, TA, and DIC though.

AR: In the revised manuscript we have reported accuracy and precision as determined from repeat analyses of CRM; For TA, precision of the data set is reported as one standard deviation determined from 56 replicate measurements of CRM Batch 154 and was 0.79 micromole per kg SW. Accuracy of TA for the data set is reported as average difference (abs(measured - known value)) between measured and known value of the same 56 replicate measurements of CRM Batch 154, and was 0.56 $\pm$ 0.55 micromole per kg SW. The average difference between 31 duplicate sample analyses was 0.76 $\pm$ 0.83 micromole per kg sw.

For DIC, precision of the data set is reported as one standard deviation determined from 49 replicate measurements of CRM Batch 154 and was 1.91 micromole per kg SW. Accuracy of DIC for the data set is reported as average difference (abs(measured - known value)) between measured and know value of the same 49 replicate measurements of CRM Batch 154 and was 1.50 $\pm$1.17micromole per kg SW. The average difference between 37 duplicate sample analyses was 1.9 $\pm$ 1.5 micromole per kg SW.

We have included this information in the revised manuscript in the methods section.

R1: What carbonate parameters are actually used for the pCO2 and saturation state calculations? This was unclear to me at this point of the manuscript but later it states they were calculated from TA-pH pairing. Please clarify in the methods.

AR: We measured all three carbonate parameters and found the calculated $\Omega$arag values similar between the DIC-pH and TA-pH pairs, not surprising given that solubility is highly pH dependent. We did however observe differences between the measured and calculated TA. Processes unrelated to calcification can impact TA values that are not accounted for in calculations but may contribute to the TA measurements. Therefore, to be conservative, we have chosen to present $\Omega$arag and pCO2 based on the DIC-pH pairs in the revised manuscript.

R1: What kind of filters were used for nutrients and carbonate chemistry sampling? Some filters may alter the values due to reactions between seawater and the material of the filters.

AR: A cellulose nitrate 0.45-$\mu$m filter and 0.20-$\mu$m polyethersulfone syringe filter were used to provide sterile sampling (i.e., low extractables) to ensure sample integrity and reduce the risk of contamination with pipetting. We have included this information in the revised manuscript.

R1: Results The results are very well-written. Just one clarification: Line 215: What range of dates were used to calculate values for the open ocean site?

AR: We reported a range of open ocean data from HOT station that was measured from 10/31/1988 to 12/9/2015 that can be accessed at http://hahana.soest.hawaii.edu/hot/products/products.html. We have included this additional information in the revised manuscript.

R1: Discussion Line 240: Respiration also occurs during the day, not just at night. Could state "net respiration" rather than just "respiration"

AR: The reviewer makes a valid point, we have revised this statement for clarity. The revised manuscript now reads "...net respiration."

R1: Lines 249-251: How can both NCP and NCC dominate? It's unclear whether the authors are trying to say they are more balanced compared to the 2nd sampling or whether they mean "net photosynthesis (+NCP)" and "net calcification (+NCC)."

AR: The reviewer brings up an important point, we have clarified this paragraph in the revised manuscript to: "To further understand the temporal variability in carbonate chemistry over the 6-d sampling period along the reef flat, diagrams of nTA versus nDIC were plotted according to Zeebe and Wolf-Gladrow, (2001), along with vectors indicating theoretical effects of the organic carbon (NCP) and inorganic carbon (NCC) cycle on seawater chemistry (Kawahata et al., 1997;Suzuki and Kawahata, 2003) (Fig

5). As presented here, NCP refers to the balance of photosynthesis and respiration, and NCC refers to the balance between calcification and dissolution (see review by Cyronak et al., 2018). Diagrams of nTA-nDIC indicate the dominance of net photosynthesis (+NCP) and net CaCO3 precipitation (+NCC) during the first sampling period (16-19 March)." R1: Lines 252-254: Please use NEC/NEP or NCC/NCP to maintain consistency with the scales used in this study. Also, please define these terms either here or in the introduction. AR: For consistency this ratio is reported as NCC:NCP in the revised manuscript. As described in the previous author response, NCP refers to the balance of photosynthesis and respiration, and NCC refers to the balance between calcification and dissolution (see review by Cyronak et al., 2018).

R1: Lines 254-255: These should be "net calcification" and "net photosynthesis" to more accurately reflect what is actually measured. NCC and NCP can indicate net processes (calcification-dissolution or photosynthesis-respiration).

AR: The reviewer brings up an important point. For clarification and accuracy, we have revised the manuscript to "net calcification" and "net photosynthesis".

R1: Line 260: The lower NCC:NCP ratio only indicates dominance by organic carbon cycling (vs inorganic carbon cycling), not which process (photosynthesis, respiration, calcification, dissolution) is actually dominating.

AR: As the reviewer points out, NCP is controlled by the organic carbon cycle (regulated by photosynthesis and respiration) whereas NCC reflects the inorganic carbon cycle, in response to CaCO3 precipitation and dissolution. The NCC:NCP ratio is defined as $1/[(2/m)-1]$ where $m$ is the slope of the nTA-nDIC plot. Therefore, changes in the NCC:NCP ratio are inferred to represent changes in the balance between the various process that influence the organic and inorganic carbon cycle. This has been reworded in the revised manuscript for clarification.

R1: Lines 260-262. This statement seems a little out of place and I'm not sure what point the authors are trying to convey. Why are the slopes in this study higher than

Heron Island? Does this reflect differences in benthic community composition, ecosystem function, or a combination?

AR: We understand the reviewer's concern regarding the comparison of our NCC:NCP ratios to those at Heron Island given potential differences in community composition, water depth, etc. Therefore, we have removed this statement from the revised manuscript.

R1: Line 262: Again, "net dissolution" and "net respiration" since actual rates are not measured using this methodology

AR: In a similar response to above, we have revised the manuscript to "net dissolution" and "net respiration" for accuracy.

R1: Does the nitrate end member at the vent site vary temporally? I appreciate using the available data to show the SGD but wonder how closely it represents discharge during the time of this study.

AR: The SGD end-member nitrate concentration was similar at both high and low tide, 117.26 and 117.13 $\mu$mol L-1 respectively, demonstrating consistency over a tidal range from water sampled directly from the vent using a piezometer inserted into the vent. However, interannual variability is possible given the range reported by Swarzenski et al. (2016) from collections in 2010 and 2013, 41.3 and 91.5 $\mu$mol L-1, respectively. However, evaluating multi-year variability is outside the scope of this present study.
R1: Figures/Figure Captions Line 389: "seep site AND on the nearshore. . .

AR: We thank the reviewer for brining to our attention this typo, the revised manuscript has been corrected accordingly.

R1: Line 393: So were TA and pH used to calculate pCO2 and saturation state? This was not clear in the methods.

AR: As described above we have chosen to present $\Omega$arag and pCO2 DICbased on the DIC-pH pairs in the revised manuscript.

R1: Figure 5: Please show error bars for the open ocean since this presumably represents a mean. NCC and NCP need to be defined either in the caption or text. In part E, these should all be shown as "net. . ." Rather than just showing the arrows for part E, could you put it on a TA/DIC plot? It can even be shown right on the plots for A-D. Given your discussion of the data, I personally would rather see the processes as small arrows on a subplot (or just in the corner of a plot) and then have dashed lines indicating the transitions between +NCC/-NCC and +NCP/-NCP. I think this would make it easier for the reader to go back and forth between the figure and discussion.

AR: Given the small error bars for the average open ocean nDIC and nTA values, we have chosen to report these values in the revised figure caption as adapted from Dore et al. (2009) since it would be difficult to view in the figure. Per the reviewer's suggestion we have revised Figure 5 and have embedded the information from Part E to Parts A-D. NCC and NCP are defined in the Discussion Section.

R1: "Technical Corrections" Line 47: no comma necessary Line 97: no comma necessary Line 112: Is 115 a typo? Should it be 15? Lines 154 and 157: parentheses just around the year Line 297: no space in SGD-driven

AR: Thank you for brining to our attention these technical corrections. We have made these corrections in the revised manuscript. However, the distance of the two deeper sites, S3 and S4 were located 115 m offshore therefore no change has been made.
* * *
[Figure]

[Figure]

**Fig. 1.** Revised Figure 5

[Figure]

---

## Author Comment (AC2) · 21 Mar 2018

Prouty et al. Anonymous Referee #2

R2: This is a very interesting and very well-written paper that will definitely be a nice contribution to the field. There are a few major and minor comments below that I feel need to be addressed prior to publication.

AR: We appreciate the valuable comments from Anonymous Referee #2 and believe we have conscientiously addressed the suggestions in the text and have detailed our responses below.

R2: My biggest criticism is that the authors did not account for TA and DIC fluxes from the SGD itself. This is an important step to interpret how much of the delta TA or delta DIC is due to reef metabolism. The authors also need to add a data analysis section to the methods and state all their statistical approaches and programs used to analyze the data. The remaining comments are relatively minor.

AR: As recommended by the reviewer, we calculated the contribution of TA and DIC from SGD at all four reef flat sites for the time period when salinity was lowest at the vent site (10.64) and the greatest contribution of SGD water likely occurred. The average residuals (calculated as the difference between the measured and non-zero salinity normalization following Richardson et al., 2018) for TA and DIC were 12±6 and 26±12 $\mu$mol kg-1, respectively. The range of TA at the reef flat sites over the course of the experiment was 706 $\mu$mol kg-1, and the range of DIC was 460 $\mu$mol kg-1. The maximum contribution from SGD (at lowest vent site salinity) could have accounted for 1.7% of the variability, and SGD DIC could only have accounted for 5.7% of DIC variability. At the S1 site, closest to the vent, the range of TA and DIC variability over the course of the experiment was 192 and 459 $\mu$mol kg-1, respectively with SGD accounting for 6.3% and 5.7% of the variability in TA and DIC, respectively.

Per the reviewer's suggestion, we have expanded the methods section to include a brief overview of the statistical methods.

R2: Line 52: There are other carbonate data for Kahekili (see, Silbiger et al. 2017 Ecology), but it is extremely limited. This is by far the most comprehensive study at this site, but "no field-based measurements" is inaccurate.

AR: Thank you for bringing this to our attention. This statement has been revised to "Building upon these studies, we present a comprehensive study to characterize the carbonate system parameters from the reefs in this area." We have also included reference to Silbiger et al. (2017) in the revised manuscript.

R2: Line 81: Change "plants" to calcifying algae

AR: Per the reviewer's suggestion "plants" has been changed to "calcifying algae". R2: Line 85: This is the first at Kahekili, but not the first to constrain carbonate chemistry in response to SGD (see Richardson et al. 2017 L&O). I would remove this sentence.

AR: We have modified this statement given previous work at Black Point, Oahu where proximal on-site sewage disposal has been identified as a nutrient source to groundwater discharge (Richardson et al., 2017). In addition, we have included this reference in the revised manuscript (Introduction Section 1).

R2: Line 124: Put both accuracy and precision of the instruments.

AR: Per the reviewer's comment, we have included both accuracy and precision in the measurements presented in Section 2.2.

"TA and DIC sample accuracy were within 0.56 $\pm$0.55 and 1.50 $\pm$ 1.17 $\mu$mol kg-1 of certified reference material respectively. Precision for TA based on replicate sample analyses was 0.76 $\pm$ 0.83 $\mu$mol kg-1. Precision for DIC based on replicate sample analyses was 1.9 $\pm$ 1.5 $\mu$mol kg-1."

R2: Line 168: Why did you use the TA-pH pairs rather than the TA-DIC pairs for the omega calculations? TA-pH is fine, but TA-DIC has less error propagation for calculating omega and it seems that you have those data.

AR: We measured all three carbonate parameters and found the calculated $\Omega$arag values similar between the DIC-pH and TA-pH pairs, not surprising given that solubility is highly pH dependent. We did however observe differences between the measured and calculated TA. Processes unrelated to calcification can impact TA values that are not accounted for in calculations but may contribute to the TA measurements. Therefore, to be conservative, we have chosen to present $\Omega$arag (and pCO2) based on the DIC-pH pairs in the revised manuscript.

R2: Line 171: It is not clear which TA, DIC values you are talking about here.

AR: For clarification, we have inserted "along the reef flat" in this statement.

R2: Add a data or statistical analysis section at the end of the methods and discuss how you analyzed your data here. What program did you use for your stats?

AR: Per the reviewer's suggestion, we have included a brief overview of the statistical methods/approach in a new section (2.4).

2.4 Statistical Analysis (new section) Slope of salinity normalized total alkalinity (nTA): salinity normalized dissolved inorganic carbon (DIC), net community calcification: net community production ratio (NCC:NCP=2$\Delta$DIC/$\Delta$TA-1) (Suzuki and Kawahata, 2003), correlation coefficients (r2), analysis of variance (ANOVA), and standard error of difference (SEdif) were calculated in Excel v. 14.7.6. Histogram plots and cubic spline fits were made in KaleidaGraph 4.1.3. As described in Section 2.3, the full seawater CO2 system was calculated using an Excel Workbook Macro translation of the original CO2SYS program (Pierrot et al., 2006).

R2: What were the TA values coming directly out of the seep?

AR: As shown in the Figure 2 and available in Prouty et al. (2017a,b) the TA values measured at the vent site ranged between 2300 to 2700 $\mu$mol kg-1.

R2: When calculating delta TA and DIC, the SGD endpoint needs to be taken into account. SGD can have a dramatically different TA and DIC concentrations than seawater (see Nelson et al. 2015 Marine Chem). A good portion of the TA and DIC fluxes are thus likely due to SGD and the remainder after accounting for these fluxes are due to bio- logical processes (e.g., calcification, dissolution, P,R). Examples of studies that have accounted for fluxes of TA and/or DIC from freshwater sources are Paquay et al 2007 Aquatic geochem or Richardson et al. 2017 L&O

AR: The reviewer is correct; SGD can dramatically impact the TA and DIC concentrations (e.g., Nelson et al., 2015), and this is clearly captured in the fact that all carbonate parameters adjacent to the primary seep site behaved conservatively with respect to salinity (Prouty et al., 2017a,b). Similarly, freshwater fluxes in a river-estuary system

can alter TA and DIC, for example Paquay et al. (2007) noted that TA and DIC in an estuary on the Big Island of Hawaii were conservative with respective to salinity. Therefore, the conservative behavior of DIC and TA with respect to salinity highlights the influence of freshwater on the carbonate chemistry system and should be accounted for in reef areas exposed to freshening from SGD (e.g., Richardson et al., 2017).

As discussed above, we calculated the contribution of TA and DIC from SGD at all four reef flat sites for the time period when salinity was lowest at the vent site (10.64) and the greatest contribution of SGD water likely occurred. The maximum contribution from SGD could have accounted for 1.7% of the variability, and SGD DIC could only have accounted for 5.7% of DIC variability. At the S1 site, closest to the vent, the range of TA and DIC variability over the course of the experiment was 192 and 459 $\mu$mol kg-1, respectively with SGD accounting for 6.3% and 5.7% of the variability in TA and DIC, respectively. We observed a very typical biotic response in the DIC and TA data, as shown in the diurnal DIC and TA plots in Figure 3 and lack of conservative behavior with respect to salinity (see new Figure S1). Adjacent to the vent site, abiotic processes, specifically SGD is driving changes in TA and DIC variability however along the reef flat biotic process dominated the TA and DIC signal.

R2: Line 234: The TA amplitude could also be indicative of high dissolution rates or a biproduct of the TA flux from the SGD onto the reef.

AR: We agree with the reviewer's comment that higher dissolution rates would drive higher TA concentrations (as well as DIC concentrations), however we only observed lower amplitude in the nTA diurnal range, rather than an increase in total concentration.

R2: Line 251: Put this information in the methods and explain how you did the calculation in addition to citing the paper.

AR: Per the reviewer's suggestion, we have expanded the methods section to include a brief overview of the statistical methods, including how we calculated the slope values of the nDIC-nTA plots.
R2: Line 290: remove "on the short term" at the end of the sentence. There is no physiology data in this study, so this sentence is a bit of a stretch. It does however look at ecosystem functioning of reefs.

AR: Per the reviewer's suggestion, we have removed the text "on the short term" in the revised manuscript.

R2: Line 297: add a citation after "environment."

AR: Per the reviewer's suggestion we have included a reference in this statement (Sunda and Cai 2012).

R2: In the discussion, it would be interesting if the authors compared their results to with other studies that also measured carbonate chemistry at SGD sites (e.g., Nelson et al. 2015 Marine Chem and Richardson et al. 2017). Are the patterns similar or different?

AR: The reviewer brings up an important point and we have expanded the manuscript to include comparisons to previously published studies, particularly those from Maunalua Bay (e.g., Nelson et al., 2015; Richardson et al., 2017). For example, the spatial gradient observed in net dissolution at sites closest to the SGD in Maunalua Bay are consistent with results from Kahekili where lower NCC:NCP ratios at the shallow sites highlights the greater vulnerability of the shallow sites to net dissolution (-NCC) under lower pH conditions relative to the deeper sites.

R2: Figures: make the colors more contrasting in the figures so that people printing in black and white can see the differences.

AR: Figures 2-5 were originally submitted as black and white and per the editor's suggestion we revised the figures to color.

Please also note the supplement to this comment:
https://www.biogeosciences-discuss.net/bg-2018-35/bg-2018-35-AC2-supplement.pdf

[Figure]

**Supplement:**

---

## Author Response (AR1)

**Editorial Comments (EC) to the Author:**
The discussion phase of your manuscript is now over and I encourage you to submit a version revised along the lines of your replies to the referees' comments. Please also consider the following comments:

**EC:** Net respiration a suggested by referee #1 does not mean anything and could generate confusion. I recommend using dark respiration which is a commonly used expression.

**AR:** We understand this comment is in reference to describing the diurnal cycle as described in Section 4.0. This statement (line 240 in original manuscript), has been revised to "DIC decreased during the day due to photosynthesis, whereas at nighttime, pH decreased and DIC increased in response to dark respiration (Fig. 3)."

**EC:** For reporting date and time, use ISO 8601 (2018-03-21)

**AR**: Per the editor's suggestion, we have revised the reporting nomenclature in the revised manuscript.

**EC** When discussing DIC-TA plots, you may find the paper by Cyronak et al. (2018; PLoS ONE) useful. I believe it was published after you submitted your manuscript.

**AR**: During our revision process we accessed the paper by Cyronak et al. (2018) and have included reference to this recently published study in the context of using carbonate parameters to evaluate reef health.

**EC**: Finally, Biogeosciences strongly promotes the full availability of the data sets reported in the papers that it publishes in order to facilitate future data comparison and compilation as well as meta-analysis. This can be achieved by uploading the data sets in an existing database and providing the link(s) in the paper. Alternatively, the data sets can be published, for free, alongside the paper as supplementary information. The ascii (or text) format is preferred for data and any format can be handled for movies, animations etc...

**AR**: We have included a statement at the end of the Acknowledgement Section: " Additional data to support this project can be found in Prouty et al. (2017b)." This reference is a USGS data release.
**R1:** "General Comments" Overall, this is a very nice paper that is scientifically sound and contains very few technical errors. The authors measured seawater carbonate chemistry and nutrients at shallow fringing reefs around a submarine groundwater discharge site to show what's driving chemical variability at these shallow sites with local anthropogenic stressors. They showed that chemistry (salinity, carbonate chemistry, DO) was highly variable at the vent site and driven by SGD while most parameters had a diurnal signal on the reef due to benthic metabolism. They also showed that areas closest to the vent site experienced a shift in NCC and NCP that may relate to nutrients being discharged from the vent. This study is scientifically sound and addresses a critical knowledge gap of understanding natural drivers of seawater carbonate chemistry variability on reefs, which must be understood in order to predict the effects of long-term anthropogenic ocean acidification on reefs. My main critique of this paper is clarification of the terminology in order to more accurately draw conclusions about benthic metabolism from the available data they collected.

**AR:** We appreciate the valuable comments from Anonymous Referee #1 and believe we have conscientiously addressed the suggestions in the text and have detailed our responses below.

**R1:** "Specific Comments" Introduction This manuscript gave a nice introduction to the re- search and sets the reader up for understanding and interpreting the results. However, the research goals were stated twice and therefore seemed repetitive. Typically, the research objectives are listed near the end of the introduction. It also was difficult to tie different parts of the introduction together, but hopefully the specific comments below will help address the flow:

**AR:** We agree with the reviewer's critique regarding repetition in the introductory paragraph and we have omitted the research objectives stated earlier in the introductory paragraph.

**R1: Lines 36-37:** Need to define OA versus coastal acidification. I assume the authors are referring to OA as a long-term anthropogenic effect owing to uptake of CO2 while coastal acidification refers to natural processes.

**AR**: As noted by the reviewer, it is important to distinguish between ocean acidification (OA) and coastal acidification. Ocean acidification is largely driven by the uptake of atmospheric anthropogenic $CO_2$ in oceanic waters (Orr et al., 2005) whereas coastal acidification is believe to be largely explained by processes such as contributions from freshwater inflow, upwelling and/or eutrophication (Cai et al., 2011) whereby excess nutrient loading from human activities to coastal waters enhances respiratory processes that release $CO_2$ and in turn increase coastal water acidity (e.g., see review by Strong et al., 2015). We have included a clarification statement in the revised manuscript in the Introduction section.

**R1: Lines 36-40**: These are nice introductions to stressors on reefs and community metabolism; however, the tie between the two is not clear as presently written. Perhaps consider adding a transition between these two statements stating how these stressors are affecting reefs (e.g. decreased calcification, increased dissolution, etc.) and then go into community metabolism.

**AR:** Per the reviewer's suggestion, we have modified these statements for greater clarity and have revised the introduction to include the following statement: "These stressors can lead to a decrease in reef health by removing grazing fish, decreasing calcification rates, and increasing nutrient and contaminant concentrations, thereby shifting the balance between reef accretion and bioerosion."

**R1: Lines 52-53:** Again, I felt like this was an abrupt transition. Could add "which may influence reef metabolism and community composition" at the end of the sentence.

**AR:** Per the reviewer's suggestions we have modified this statement to include "…which may influence reef metabolism and community composition by changing coastal water quality."

**R1: Line 57**: add "calcium carbonate (CaCO3)" in front of dissolution

**AR:** Per the reviewer's suggestions we have inserted "calcium carbonate (CaCO3)" in front of dissolution

**R1: Methods Lines 95-96:** What is the other 90% of cover where there is 10% live coral cover? What is the community composition of the other 49% of hard-bottom area? This would help with interpretation of results and DIC/TA slopes as this relates to the community composition (corals vs algae vs sand, etc. See Page et al 2016 for reference on community composition influence on seawater carbonate chemistry.)

**AR:** A detailed discussion of seafloor-bottom type can be found in Cochran et al. (2014). In brief, the remaining 49% of available hardbottom consists of aggregate reef, spur-and-groove, patch reefs, pavement, and reef rubble, which as the reviewer points can influence seawater carbonate chemistry (Page et al., 2017). In addition to live coral cover, Cochran et al. (2014) observed macroalgae, corraline algae, seagrass, and turf in the area mapped (5 km$^2$ of sea floor from the shoreline to water depths of ~30 m), however the sampling sites in our study were areas of live coral cover. We have included additional information in the Methods Section and reference to Page et al. (2017) in the Introduction Section.

**R1: Lines 111-114:** What was the approximate depth of the vent site? This would be valuable information in interpreting the variability (measured as daily range) of chemistry since depth can be such a strong control (Falter et al 2013).

**AR:** The vent site was located at a comparable depth to the two shallow (<1.5 m) sites (S1 and S2). We have inserted the water depth in the revised manuscript.

**R1:** I do wonder about any algae, bacterial films, etc. that may have grown on the inside of the tubing and possibly influenced carbonate chemistry and nutrients. Were there any tests (e.g. sampling carbonate chemistry near the intake and at the outtake) to assess whether the tubing was clean throughout the entire field study?

**AR**: Sampling tubes were flushed for a minimum of 20 minutes to remove residual seawater before collecting data and water samples. In addition, the tube intakes were fitted with a stainless steel screen cap to prevent uptake of large particulates. We also inspected the tubes upon extraction and found no significant algal growth. The revised manuscript includes this additional information.

**R1:** Thanks for providing the approximate precision of the TA and DIC measurements. It would be great to see the actual precision and accuracy (as mean plus/minus sd) of pH, TA, and DIC though.

**AR**: In the revised manuscript we have reported accuracy and precision as determined from repeat analyses of CRM; For TA, precision of the data set is reported as one standard deviation determined from 56 replicate measurements of CRM Batch 154 and was 0.79 micromole per kg SW. Accuracy of TA for the data set is reported as average difference (abs(measured - known value)) between measured and known value of the same 56 replicate measurements of CRM Batch 154, and was $0.56 \pm 0.55$ micromole per kg SW. The average difference between 31 duplicate sample analyses was $0.76 \pm 0.83$ micromole per kg sw.

For DIC, precision of the data set is reported as one standard deviation determined from 49 replicate measurements of CRM Batch 154 and was 1.91 micromole per kg SW. Accuracy of DIC for the data set is reported as average difference (abs(measured - known value)) between measured and know value of the same 49 replicate measurements of CRM Batch 154 and was 1.50 ±1.17micromole per kg SW. The average difference between 37 duplicate sample analyses was 1.9 ± 1.5 micromole per kg SW.

We have included this information in the revised manuscript in the methods section.

**R1:** What carbonate parameters are actually used for the pCO2 and saturation state calculations? This was unclear to me at this point of the manuscript but later it states they were calculated from TA-pH pairing. Please clarify in the methods.

**AR:** We measured all three carbonate parameters and found the calculated $\Omega_{arag}$ values similar between the DIC-pH and TA-pH pairs, not surprising given that solubility is highly pH dependent. We did however observe differences between the measured and calculated TA. Processes unrelated to calcification can impact TA values that are not accounted for in calculations but may contribute to the TA measurements. Therefore, to be conservative, we have chosen to present $\Omega_{arag}$ and $pCO_2$ based on the DIC-pH pairs in the revised manuscript.

R1: What kind of filters were used for nutrients and carbonate chemistry sampling? Some filters may alter the values due to reactions between seawater and the material of the filters.

AR: A cellulose nitrate 0.45-µm filter and 0.20-µm polyethersulfone syringe filter were used to provide sterile sampling (i.e., low extractables) to ensure sample integrity and reduce the risk of contamination with pipetting. We have included this information in the revised manuscript.

**R1:** Results The results are very well-written. Just one clarification:
**Line 215:** What range of dates were used to calculate values for the open ocean site?

**AR:** We reported a range of open ocean data from HOT station that was measured from 10/31/1988 to 12/9/2015 that can be accessed at http://hahana.soest.hawaii.edu/hot/products/products.html. We have included this additional information in the revised manuscript.

**R1:** Discussion Line 240: Respiration also occurs during the day, not just at night. Could state "net respiration" rather than just "respiration"

**AR:** We have revised this statement for clarity. The revised manuscript now reads "…dark respiration".

**R1: Lines 249-251:** How can both NCP and NCC dominate? It's unclear whether the authors are trying to say they are more balanced compared to the 2nd sampling or whether they mean "net photosynthesis (+NCP)" and "net calcification (+NCC)."

**AR:** The reviewer brings up an important point, we have clarified this paragraph in the revised manuscript to: "To further understand the temporal variability in carbonate chemistry over the 6-d sampling period along the reef flat, diagrams of $n$TA versus $n$DIC were plotted according to Zeebe and Wolf-Gladrow, (2001), along with vectors indicating theoretical effects of the organic carbon (NCP) and inorganic carbon (NCC) cycle on seawater chemistry (Kawahata et al., 1997;Suzuki and Kawahata, 2003) (Fig 5). As presented here, NCP refers to the balance of photosynthesis and respiration, and NCC refers to the balance between calcification and dissolution (see review by Cyronak et al., 2018). Diagrams of $n$TA-$n$DIC indicate the dominance of net photosynthesis (+NCP) and net CaCO3 precipitation (+NCC) during the first sampling period (16-19 March)."

**R1: Lines 252-254:** Please use NEC/NEP or NCC/NCP to maintain consistency with the scales used in this study. Also, please define these terms either here or in the introduction.

**AR:** For consistency this ratio is reported as NCC:NCP in the revised manuscript. As described in the previous author response, NCP refers to the balance of photosynthesis and respiration, and NCC refers to the balance between calcification and dissolution (see review by Cyronak et al., 2018).

**R1: Lines 254-255:** These should be "net calcification" and "net photosynthesis" to more accurately reflect what is actually measured. NCC and NCP can indicate net processes (calcification-dissolution or photosynthesis-respiration).

**AR:** The reviewer brings up an important point. For clarification and accuracy, we have revised the manuscript to "net calcification" and "net photosynthesis".

**R1: Line 260:** The lower NCC:NCP ratio only indicates dominance by organic carbon cycling (vs inorganic carbon cycling), not which process (photosynthesis, respiration, calcification, dissolution) is actually dominating.

**AR**: As the reviewer points out, NCP is controlled by the organic carbon cycle (regulated by photosynthesis and respiration) whereas NCC reflects the inorganic carbon cycle, in response to $CaCO_3$ precipitation and dissolution. The NCC:NCP ratio is defined as $1/[(2/m)-1]$ where m is the slope of the $n$TA-$n$DIC plot. Therefore, changes in the NCC:NCP ratio are inferred to represent changes in the balance between the various process that influence the organic and inorganic carbon cycle. This has been reworded in the revised manuscript for clarification.

**R1: Lines 260-262.** This statement seems a little out of place and I'm not sure what point the authors are trying to convey. Why are the slopes in this study higher than Heron Island? Does this reflect differences in benthic community composition, ecosystem function, or a combination?

**AR:** We understand the reviewer's concern regarding the comparison of our NCC:NCP ratios to those at Heron Island given potential differences in community composition, water depth, etc. Therefore, we have removed this statement from the revised manuscript.

**R1: Line 262:** Again, "net dissolution" and "net respiration" since actual rates are not measured using this methodology

**AR:** We have revised the manuscript to "net dissolution" and "net respiration" for greater clarity.

**R1:** Does the nitrate end member at the vent site vary temporally? I appreciate using the available data to show the SGD but wonder how closely it represents discharge during the time of this study.

**AR:** The SGD end-member nitrate concentration was similar at both high and low tide, 117.26 and 117.13 µmol L$^{-1}$ respectively, demonstrating consistency over a tidal range from water sampled directly from the vent using a piezometer inserted into the vent. However, interannual variability is possible given the range reported by Swarzenski et al. (2016) from collections in 2010 and 2013, 41.3 and 91.5 µmol L$^{-1}$, respectively. However, evaluating multi-year variability is outside the scope of this present study.

**R1:** Figures/Figure Captions Line 389: "seep site AND on the nearshore. . .

**AR:** We thank the reviewer for brining to our attention this typo, the revised manuscript has been corrected accordingly.

**R1: Line 393**: So were TA and pH used to calculate pCO2 and saturation state? This was not clear in the methods.

**AR:** As described above we have chosen to present $\Omega_{arag\ and}$ pCO2 DICbased on the DIC-pH pairs in the revised manuscript.

**R1:** Figure 5: Please show error bars for the open ocean since this presumably represents a mean. NCC and NCP need to be defined either in the caption or text. In part E, these should all be shown as "net. . ." Rather than just showing the arrows for part E, could you put it on a TA/DIC plot? It can even be shown right on the plots for A-D. Given your discussion of the data, I personally would rather see the processes as small arrows on a subplot (or just in the corner of a plot) and then have dashed lines indicating the transitions between +NCC/-NCC and +NCP/-NCP. I think this would make it easier for the reader to go back and forth between the figure and discussion.

**AR:** Given the small error bars for the average open ocean $n$DIC and $n$TA values, we have chosen to report these values in the revised figure caption as adapted from Dore et al. (2009) since it would be difficult to view in the figure. Per the reviewer's suggestion we have revised Figure 5 and have embedded the information from Part E to Parts A-D.

NCC and NCP are defined in the Discussion Section.

**R1:** "Technical Corrections" Line 47: no comma necessary Line 97: no comma necessary Line 112: Is 115 a typo? Should it be 15? Lines 154 and 157: parentheses just around the year Line 297: no space in SGD-driven

**AR:** Thank you for brining to our attention these technical corrections. We have made these corrections in the revised manuscript. However, the distance of the two deeper sites, S3 and S4 were located 115 m offshore therefore no change has been made.
**R2:** This is a very interesting and very well-written paper that will definitely be a nice contribution to the field. There are a few major and minor comments below that I feel need to be addressed prior to publication.

**AR:** We appreciate the valuable comments from Anonymous Referee #2 and believe we have conscientiously addressed the suggestions in the text and have detailed our responses below.

**R2**: My biggest criticism is that the authors did not account for TA and DIC fluxes from the SGD itself. This is an important step to interpret how much of the delta TA or delta DIC is due to reef metabolism. The authors also need to add a data analysis section to the methods and state all their statistical approaches and programs used to analyze the data. The remaining comments are relatively minor.

**AR**: As recommended by the reviewer, we calculated the contribution of TA and DIC from SGD at all four reef flat sites for the time period when salinity was lowest at the vent site (10.64) and the greatest contribution of SGD water likely occurred. The average residuals (calculated as the difference between the measured and non-zero salinity normalization following Richardson et al., 2018) for TA and DIC were $12\pm6$ and $26\pm12$ $\mu$mol kg$^{-1}$, respectively. The range of TA at the reef flat sites over the course of the experiment was 706 $\mu$mol kg$^{-1}$, and the range of DIC was 460 $\mu$mol kg$^{-1}$. The maximum contribution from SGD (at lowest vent site salinity) could have accounted for 1.7% of the variability, and SGD DIC could only have accounted for 5.7% of DIC variability. At the S1 site, closest to the vent, the range of TA and DIC variability over the course of the experiment was 192 and 459 $\mu$mol kg$^{-1}$, respectively with SGD accounting for 6.3% and 5.7% of the variability in TA and DIC, respectively.

Per the reviewer's suggestion, we have expanded the methods section to include a brief overview of the statistical methods.

**R2**: Line 52: There are other carbonate data for Kahekili (see, Silbiger et al. 2017 Ecology), but it is extremely limited. This is by far the most comprehensive study at this site, but "no field-based measurements" is inaccurate.

**AR:** Thank you for bringing this to our attention. This statement has been revised to "Building upon these studies, we present a comprehensive study to characterize the carbonate system parameters from the reefs in this area." We have also included reference to Silbiger et al. (2017) in the revised manuscript.

**R2:** Line 81: Change "plants" to calcifying algae

**AR:** Per the reviewer's suggestion "plants" has been changed to "calcifying algae".

R2: Line 85: This is the first at Kahekili, but not the first to constrain carbonate chemistry in response to SGD (see Richardson et al. 2017 L&O). I would remove this sentence.

AR: We have modified this statement given previous work at Black Point, Oahu where proximal on-site sewage disposal has been identified as a nutrient source to groundwater discharge (Richardson et al., 2017). In addition, we have included this reference in the revised manuscript (Introduction Section 1).

**R2:** Line 124: Put both accuracy and precision of the instruments.

**AR:** Per the reviewer's comment, we have included both accuracy and precision in the measurements presented in Section 2.2.

**R2**: Line 168: Why did you use the TA-pH pairs rather than the TA-DIC pairs for the omega calculations? TA-pH is fine, but TA-DIC has less error propagation for calculating omega and it seems that you have those data.

**AR:** We measured all three carbonate parameters and found the calculated $\Omega_{arag}$ values similar between the DIC-pH and TA-pH pairs, not surprising given that solubility is highly pH dependent. We did however observe differences between the measured and calculated TA. Processes unrelated to calcification can impact TA values that are not accounted for in calculations but may contribute to the TA measurements. Therefore, to be conservative, we have chosen to present $\Omega_{arag}$ (and $pCO_2$) based on the DIC-pH pairs in the revised manuscript.

**R2:** Line 171: It is not clear which TA, DIC values you are talking about here.

**AR:** For clarification, we have inserted "along the reef flat" in this statement.

**R2:** Add a data or statistical analysis section at the end of the methods and discuss how you analyzed your data here. What program did you use for your stats?

**AR:** Per the reviewer's suggestion, we have included a brief overview of the statistical methods/approach in a new section (2.4).

**R2:** What were the TA values coming directly out of the seep?

**AR:** As shown in the Figure 2 and available in Prouty et al. (2017a,b) the TA values measured at the vent site ranged between 2300 to 2700 $\mu$mol kg$^{-1}$.

**R2:** When calculating delta TA and DIC, the SGD endpoint needs to be taken into account. SGD can have a dramatically different TA and DIC concentrations than seawater (see Nelson et al. 2015 Marine Chem). A good portion of the TA and DIC fluxes are thus likely due to SGD and the remainder after accounting for these fluxes are due to bio- logical processes (e.g., calcification, dissolution, P,R). Examples of studies that have accounted for fluxes of TA and/or DIC from freshwater sources are Paquay et al 2007 Aquatic geochem or Richardson et al. 2017 L&O

**AR:** The reviewer is correct; SGD can dramatically impact the TA and DIC concentrations (e.g., Nelson et al., 2015), and this is clearly captured in the fact that all carbonate parameters adjacent to the primary seep site behaved conservatively with respect to salinity (Prouty et al., 2017a,b). Similarly, freshwater fluxes in a river-estuary system can alter TA and DIC, for example Paquay et al. (2007) noted that TA and DIC in an estuary on the Big Island of Hawaii were conservative with respective to salinity. Therefore, the conservative behavior of DIC and TA with respect to salinity highlights the influence of freshwater on the carbonate chemistry system and should be accounted for in reef areas exposed to freshening from SGD (e.g., Richardson et al., 2017).

As discussed above, we calculated the contribution of TA and DIC from SGD at all four reef flat sites for the time period when salinity was lowest at the vent site (10.64) and the greatest contribution of SGD water likely occurred. The maximum contribution from SGD could have accounted for 1.7% of the variability, and SGD DIC could only have accounted for 5.7% of DIC variability. At the S1 site, closest to the vent, the range of TA and DIC variability over the course of the experiment was 192 and 459 $\mu$mol kg$^{-1}$, respectively with SGD accounting for 6.3% and 5.7% of the variability in TA and DIC, respectively.

We observed a very typical biotic response in the DIC and TA data, as shown in the diurnal DIC and TA plots in Figure 3 and lack of conservative behavior with respect to salinity (see new Figure S1).

Adjacent to the vent site, abiotic processes, specifically SGD is driving changes in TA and DIC variability however along the reef flat biotic process dominated the TA and DIC signal.

**R2:** Line 234: The TA amplitude could also be indicative of high dissolution rates or a biproduct of the TA flux from the SGD onto the reef.

**AR:** We agree with the reviewer's comment that higher dissolution rates would drive higher TA concentrations (as well as DIC concentrations), however we only observed lower amplitude in the $n$TA diurnal range, rather than an increase in total concentration.

**R2:** Line 251: Put this information in the methods and explain how you did the calculation in addition to citing the paper.

**AR:** Per the reviewer's suggestion, we have expanded the methods section to include a brief overview of the statistical methods, including how we calculated the slope values of the $n$DIC-$n$TA plots.

**R2:** Line 290: remove "on the short term" at the end of the sentence. There is no physiology data in this study, so this sentence is a bit of a stretch. It does however look at ecosystem functioning of reefs.

**AR:** Per the reviewer's suggestion, we have removed the text "on the short term" in the revised manuscript.

**R2:** Line 297: add a citation after "environment."

**AR:** Per the reviewer's suggestion we have included a reference in this statement (Sunda and Cai 2012).

**R2:** In the discussion, it would be interesting if the authors compared their results to with other studies that also measured carbonate chemistry at SGD sites (e.g., Nelson et al. 2015 Marine Chem and Richardson et al. 2017). Are the patterns similar or different?

**AR:** The reviewer brings up an important point and we have expanded the manuscript to include comparisons to previously published studies, particularly those from Maunalua Bay (e.g., Nelson et al., 2015; Richardson et al., 2017). For example, the spatial gradient observed in net dissolution at sites closest to the SGD in Maunalua Bay are consistent with results from Kahekili where lower NCC:NCP ratios at the shallow sites highlights the greater vulnerability of the shallow sites to net dissolution (-NCC) under lower pH conditions relative to the deeper sites.

**R2**: Figures: make the colors more contrasting in the figures so that people printing in black and white can see the differences.

**AR**:  Figures 2-5 were originally submitted as black and white and per the editor's suggestion we revised the figures to color.

[revised manuscript text omitted]

Nancy Prouty 1/31/2018 3:44 PM

Nancy Prouty 1/31/2018 4:07 PM

Nancy Prouty 1/31/2018 4:04 PM

Nancy Prouty 1/31/2018 3:49 PM

Nancy Prouty 1/31/2018 3:49 PM

Nancy Prouty 1/31/2018 4:04 PM

Nancy Prouty 1/31/2018 4:05 PM

of the $n$DIC-$n$TA plots were used to calculate ratios of NCC:NCP (Table 1) using methods of Suzuki and Kawahata (2003) to estimate the relative contribution of these processes to reef biogeochemistry. In the absence of reliable water mass residence time, ratios were used rather than metabolic rates. The NCC:NCP ratios for the first sampling period ranged from 0.50 to 0.87 indicating a dominance of NCP relative to NCC. Plots of $n$DIC-$n$TA (Fig. 5) indicate that these sites were dominated primary by net photosynthesis and net calcification. This pattern was observed at all four sites along the reef flat. The lower NCC:NCP ratios at the shallow sites highlight the greater vulnerability of the shallow sites to net dissolution (-NCC) under lower pH conditions relative to the deeper sites. These results are in agreement with Richardson et al. (2017) that found net dissolution at reef sites closet to groundwater vents in Maunalua Bay, Oahu. A shift occurred at all sampling sites after the first sampling period. Elevated $n$DIC and $n$TA values from 2016-03-21 to 2016-03-22 indicate a shift to respiration and net dissolution in the $n$TA-$n$DIC diagrams (Fig. 5). At the shallow sites, S1 and S2 (Fig. 5A and B), the NCC:NCP ratios were 0.56 and 0.39 during the second sampling period (Table 1), respectively, indicating the dominance of NCP relative to NCC. Net dissolution and net respiration contributed nearly equally with NCC:NCP ratios near 1.0 during the second sampling at sites S3 and S4 located further offshore. Given the salinity range along the reef flat (34 to 36), traditional salinity normalization (e.g., Friis et al., 2003) could potentially overestimate the $n$DIC and $n$TA concentrations by ~20 to ~10 µmol kg$^{-1}$ respectively, according to non-zero normalization described in Richardson et al. (2017). However, rather than reflecting an artifact of the salinity normalization, given the non-linear relation of DIC and TA to salinity along the reef flat (Fig. S1), this shift is interpreted as a reef community response. As shown in Figures 4 and 5, this change captures a shift from a reef community dominated by net calcification and net photosynthesis to one dominated by net respiration and net dissolution.

The shift from net photosynthesis (P) to net respiration (R) as captured in the $\Delta n$DIC histogram plots (Fig. 4), suggests that the coral-algal association consumed more energy than it produced during the second sampling period. As a proxy for autotrophic capacity, the change in P:R ratio may reflect an increase in coral heterotrophic feeding relative to autotrophic feeding (Coles and Jokiel, 1977;Hughes and Grottoli, 2013). Typically, stored lipid reserves in the tissue are utilized when the stable symbiotic environment is disturbed (e.g., Szmant and Gassman, 1990; Ainsworth et al., 2008). Although short-lived, thermally-induced bleaching has been linked to depletion of coral lipid reserves (e.g., Hughes and Grottoli, 2013), excess nutrient loading can also shift the stability of the coral-algae symbiosis, thereby reducing stored tissue reserves (Wooldridge, 2016). According to Glenn et al. (2013), up to 11

Nancy Prouty 1/31/2018 4:20 PM

Nancy Prouty 1/31/2018 4:18 PM

Nancy Prouty 2/27/2018 1:52 PM

Nancy Prouty 3/21/2018 2:59 PM

Nancy Prouty 3/21/2018 2:17 PM

Nancy Prouty 3/21/2018 2:17 PM

Nancy Prouty 1/31/2018 4:43 PM

Nancy Prouty 1/30/2018 6:17 PM

Nancy Prouty 1/31/2018 5:27 PM

Nancy Prouty 1/30/2018 4:09 PM

Nancy Prouty 3/21/2018 2:56 PM

Nancy Prouty 2/23/2018 4:21 PM

Nancy Prouty 2/23/2018 4:22 PM

[revised manuscript text omitted]

Nancy Prouty 3/21/2018 2:13 PM

Nancy Prouty 3/21/2018 2:14 PM